# A Study on Technology Acceptance of Digital Healthcare among Older Korean Adults Using Extended Tam (Extended Technology Acceptance Model)

Khin Shoon Lei Thant Zin [1,†], Seieun Kim [1,†], Hak-Seon Kim [2,3] and Israel Fisseha Feyissa [4,5,*]

1   Department of Global Business, Kyungsung University, Busan 48434, Republic of Korea
2   School of Hospitality and Tourism Management, Kyungsung University, Busan 48434, Republic of Korea
3   Wellness & Tourism Big Data Research Institute, Kyungsung University, Busan 48434, Republic of Korea
4   School of Global Studies, Kyungsung University, Busan 48434, Republic of Korea
5   Global Migration Research Center, Kyungsung University, Busan 48434, Republic of Korea
*   Correspondence: israel12@kyungsung.ac.kr
†   These co-first authors contributed equally to this work.

**Abstract:** The use of digital health and wrist-worn wearable technologies have been increasingly utilized, especially during COVID-19 surge, to help monitor patients and vulnerable groups such as elderly people. As one of the countries with highest aging population, South Korean older adults are expected to be familiarized with these healthcare technologies. However, there have been a few studies on the investigation of Korean older adults' attitude towards the acceptance of wearable technologies, such as a smart health watch after the COVID-19 curve flattened in South Korea. Thus, the purpose of this study is to investigate the acceptability of digital health wearable technology in healthcare by the Korean older adults and their attitude towards the use of smart health watches by using an extended Technology Acceptance Model while considering the context of the COVID-19 pandemic. We performed a cross-sectional survey of Korean adults aged 56 years and older who are living in Busan, and a total of 170 respondents were received. Results reveal that perceived usefulness, perceived ease of use, and facilitating conditions have a significant impact on older Korean's attitudes towards the use of a smart health watch, while the relationship between social influence and attitude towards its use was found to not be statistically significant. The attitude towards the use of smart health watches had an effect on their intention to use the smartwatch. By using the findings from the study, the digital wearables providers, manufacturers, and promotors can enhance their strategy to elevate the use of digital healthcare wearables among Korean elderly people while ensuring these products are of good quality and affordable, as well as ensuring necessary assistance is provided to the elderly people when utilizing and adopting these wearables in their everyday lives. Moreover, the results of this study can be utilized to accommodate the needs of Korean elderly people regarding their use of smart health watches and help promote the benefits of healthcare wearable technologies after the pandemic subsides.

**Keywords:** digital health technology; technology acceptance; health smart watch; South Korean; older adults

## 1. Introduction

According to the data reported by Statistics Korea (2021), 16.5% of South Korea's total population was above the age of 65 in 2021; it is also predicted that by 2050, the Korean aging population will occupy 43.9% of country's population. At the same time, the percent of elderly Korean people living alone is on the rise. In 2020 alone, one-person households of the Korean elderly accounted for 35.1% of the total elderly households (Statistics Korea 2021).

An increase of single households and a break of family relations are reported to be prominent reasons for lonely death cases in Korea. A solitary death or lonely death is defined as a case where a single household without any interaction with family or neighbors dies alone. Generally, it refers to a case where a dead single household is discovered after being left at home for more than a week (Choi 2018). This shows that elderly people living alone in single households without adequate healthcare provisions may further lead to a higher possibility of the elderly dying alone. Moreover, in this era of aging, it is difficult for the family alone to take the responsibility for elderly care.

As the world population grows older, the dependency of elderly individuals will also grow if senior citizens are not given the support that they need to perform their daily tasks on their own (Guner and Acarturk 2020). In that regard, Guner and Acarturk (2020) suggested that the use of information and communication technologies (ICT) by senior adults may have the potential to improve their quality of life by strengthening their ties to their families, increasing their independence, and promoting their inclusion in society. For a country with a rapidly aging population, the occurrence of lonely deaths among the aging population is one of the major social problems in South Korea that needs to be addressed and handled efficiently. Economic support and health maintenance are found to be the two most significant concerns for the Korean elderly population and many of these people have chronic diseases and age-related illnesses demanding specialized intensive care (Eun 2008).

To reduce health costs and promote self-health management by the elderly living alone, effective use of technologies and digital health technologies became necessary. The use of technology in healthcare can help increase and support the well-being of the elderly at home while promoting their independency through advances in telehealth, distance monitoring, and other technology tools (Mostaghel and Oghazi 2017). With the help of advanced technologies and digital healthcare, the accessibility of healthcare has shifted from a doctor-centered approach to a patient-centered approach (Jo et al. 2021b) In addition, the active use of digital technology by older adults will directly promote healthy aging, which will have a positive effect not only on the older adults themselves but also on society as a whole, which is rapidly aging (Mitzner et al. 2010).

In addition, due to the physical distancing and social interaction restrictions imposed by the pandemic, digital health technologies have been deployed effectively to provide medical services and healthcare for easier healthcare accessibility. The surge of COVID-19 has changed the methods for providing healthcare services and highlighted the need to monitor health conditions remotely (Abrashkin et al. 2021). Digital healthcare and technologies have become the most efficient solutions as a way of providing healthcare services to the rapidly growing elderly people while mitigating the risk of COVID-19 and the burden of frequent hospital visits.

Several technologies and digital systems have been implemented to offer more comprehensible digital healthcare and services to the elderly, such as IoT-based wearable devices, mobile health, telemedicine, telehealth, big data predictive analysis, assistive technology (AT) services, and artificial intelligence. These digital health technologies have helped in mitigating the risk and effects of COVID-19, as well as enhancing the capabilities of the elderly to independently monitor and manage their health. (Jo et al. 2021a; Vidal-Alaball et al. 2020; Abrashkin et al. 2021; Cardozo and Steinberg 2010; Hur and Park 2020).

The application of mobile health (m-health) during the pandemic has shown positive effects on telemedicine, and telehealth implementation during the COVID-19 pandemic has helped the patient care processes and increased the monitoring of patient health remotely (Abdul Rahman et al. 2022). The positive outlook and acceptance among the Korean population have increased regarding telehealth, indicating the potential use of telehealth even after the pandemic, although telehealth was allowed temporarily during the pandemic (An et al. 2021).

Despite rapid advancement in healthcare technologies, older adults often face challenges and barriers due to physical impairments (poor vision or lower motor control), cognitive difficulties (memory and speed of processing), lack of technological skills, and

a lack of perceived ability and time (Wildenbos et al. 2018; Czaja et al. 2006; Ahmad et al. 2022). The level of digital and eHealth literacy of the seniors can determine and have an impact on their level of willingness to use ICT-based technologies in elderly care (Kapadia et al. 2015). Moreover, older adults expressed that the prevalent barriers to their e-health engagement were a lack of self-efficacy, knowledge, support, functionality, and information provisions about the benefits of e-health (Wilson et al. 2021)

Regardless of advanced technology healthcare wearables, it is far more important to investigate the perspectives and attitudes of elderly people toward these new technologies and their intention to use technology-based healthcare devices. It's also essential to identify their barriers to adopting new digital healthcare technologies. The willingness of the elderly and their perception play important roles when adopting technology-based monitoring and utilizing such devices.

In order to adopt new technologies by the seniors, the level of usefulness and usability of technology need to be greater than their feeling of inadequacy, and personal, psychological, and physical contextual factors are also required to be considered in understanding the elderly's perception and their intention to use technologies (Heinz et al. 2013; Peek et al. 2016). Increased accessibility, enhanced care, usefulness, ease of use, and privacy/discomfort are found to be significant factors in telehealth acceptance by the Korean general population (An et al. 2021). Older Korean adults who have higher levels of empowerment are more likely to perceive technology use as easier and, therefore, more useful, resulting in positive attitudes toward using mobile applications (An et al. 2018). Moreover, the Korean elderly expressed their willingness to adopt an integrated smart home system (ISHS) after they acquired awareness about the benefits of IoT-based ISHS (Jo et al. 2021b). In a study by Ahmad et al. 2020, older diabetic patients' continued intention to utilize digital health wearables was investigated by extending Technology Acceptance Model (TAM) with additional components, such as perceived irreplaceability, perceived credibility, compatibility, and social influence. This study examined the factors that affect elderly diabetic patients' intention to continue using digital health wearables in the setting of Bangladesh.

Although there are some studies that investigate older adults' perception and technology acceptance in healthcare (Holden and Karsh 2010; Jo et al. 2021b; Rahimi et al. 2018; Steele et al. 2009; Vassli and Farshchian 2018; Yoo et al. 2020), there are still limited studies on the change in the acceptance and adoption of new technologies in healthcare by the elderly during the COVID-19 pandemic era. Thus, the purpose of this study is to investigate the acceptability of digital technology in healthcare by Korean older adults and their attitude toward the use of digital healthcare considering the context of the COVID-19 pandemic.

The following sections of this study were structured as follows: In Section 2, previous studies and relevant research papers were provided regarding the perception and acceptance of the elderly as well as the technology acceptance model for the development of the hypothesis. Section 3 explained the major procedures of the research methodology. Section 4 elaborated on important analytical results with explanatory tables. In Sections 5 and 6, the conclusion and discussion were provided along with the limitations of the study and direction for future studies.

## 2. Literature Review

### 2.1. Elderly Perception and Their Acceptability of Technology Use in Healthcare

Over the years, the acceptance of healthcare technologies by seniors and factors influencing their intention to use new healthcare technologies have been widely studied using different acceptance models. Using TAM (Technology Acceptance Model), (Hong et al. 2020) it was reported that the perceived usefulness has a significant influence on Korean elderly's intention to use of IoT HealthCare services. Steele et al. (2009) studied that acceptance of WSN-based systems by older adults is mainly affected by the cost and

lack of confidence in interacting with such systems, but they expressed a willingness to get training in order to enhance self-efficacy.

According to the study of Ha and Park (2020), willingness and good technology acceptance in healthcare are displayed among older Korean adults with multiple chronic conditions. Moreover, their age and education level have a high influence on their level of acceptance of technology. Gerontechnology self-efficacy and facilitating conditions were positively associated with technology acceptance among older Korean adults with multiple chronic conditions. It is reported that usefulness and ease of use had positive significant effects on the Korean elderly's intention to use digital health devices, in which perceived usefulness, self-efficacy, and anxiety had significant effects on the ease of use while self-efficacy, facilitating conditions, attitude to life, and satisfaction had significant effects on perceived usefulness (Shin et al. 2020).

Perceived value, attitude, perceived behavior control, technology anxiety, and self-actualization needs are found to be positively affected by the behavior intention of Chinese older citizens in adopting mobile health services (Deng et al. 2014). In addition, performance expectancy, effort expectancy, social influence, technology anxiety, and resistance to change had a significant effect on the users' behavioral intentions to adopt mHealth services among Bangladesh's elderly population. (Hoque and Sorwar 2017). It is identified that perceived usefulness, compatibility, facilitating conditions, and self-reported health status significantly and positively affect older adults' intention to use smart wearable systems (Li et al. 2019)

Some studies have reported and suggested the important factors for Korean elderly people in adopting technology in healthcare. Jo et al. (2021b) suggested that for better and easier adoption of new technologies and Integrated Smart Home System among the Korean elderly, sufficient awareness still needs to be raised among the elders regarding the benefits of such technologies and elderly-friendly smart home sensors are required to minimize negative responses.

Moreover, Chung et al. (2017) reported that the role of the government is important for Korean elderly in expanding the adoption and use of new technologies. Contextual factors also need to be taken into consideration when investigating the perception and attitudes of home-based monitoring technologies among elderly people from different cultural and ethnic backgrounds (Kim and Choi 2019), suggesting that healthcare technology and relevant services, particularly in public health in Korea, need to be designed and developed considering the privacy concerns and diversity among older adults, especially older women.

Digital healthcare wearable technologies have become more advanced and applied in various aspects of healthcare. Wrist-worn wearable technologies such as health smart-watches are designed to help the elderly to age successfully by monitoring their health and fitness-related data such as heart rate, blood pressure, and fall detection. Moreover, digital healthcare wearables have been utilized to alert patients to take their medications at the correct time and as an application for detection of neurological disorders and early symptoms in Parkinson's Disease in elderly patients (Lazaro et al. 2020; Gordon 2018). Sine the potential application of wearable technology is increasing, it is important to evaluate the acceptance of these technologies among Korean older adults as well as their behavioral intention to use these wearable devices.

### 2.2. Theoretical Framework: Technology Acceptance Model

There have been several competing models and theories that are used widely to explain technology acceptance. The Technology Acceptance Model ("TAM") is the leading theory in healthcare technology acceptance analyses which was designed to clarify and predict how well a person would accept IT. It has evolved to become a key model in understanding the predictors of human behavior toward potential acceptance or rejection of technology. TAM was proposed by Davis in 1989 to examine the intention of using various types of technologies and it explains the mutual relationship between external variables that

influence a user's acceptance of technology and factors that influence actual behavior to use. This model is derived from the Theory of Reasoned Action (TRA) proposed by Fishbein and Ajzen (1991). The Theory of Planned Behavior (TPB) was proposed by Ajzen (1991). According to TRA, social behavior is driven by an individual's attitude toward engaging in that behavior as well as one's beliefs regarding the outcomes and the value in those outcomes of performing those behaviors.

The model "TAM" involves two main constructs, known as "perceived usefulness (PU)" and "perceived ease of use (PEOU)," which are impacted by external variables. These two constructs affect "attitude toward using (ATT)" and "behavioral intention to use (BI)" in which BI is influenced by one's attitude toward using the IT. Davis defined perceived usefulness as "the degree to which a person believes that using a particular system would enhance his or her job performance" and perceived ease of use as "the degree to which a person believes that using a particular system would be free of effort." Users' beliefs determine their attitudes toward using the system. The model presumes a mediating role of perceived ease of use and usefulness in association with external variables and actual system use. Many previous researchers have demonstrated the validity of the model "TAM" across a wide variety of fields and disciplines (Kim and Kim 2016). Moreover, the elderly technology acceptance and adoption has been extensively researched using TAM in a range of technological contexts such as healthcare and assistive technology, social networking, online shopping, Internet, computer, online public service, and entertainment (Yap et al. 2022).

TAM (Technology Acceptance Model) is known as one of the most popular research models to predict a user's intention to perform a particular behavior and one's acceptance of information systems and technology. Depending on the context and field of study, new factors and separate variables have been added and integrated in order to develop contextualized TAM versions. This allows the enhancement of dimensions of the TAM in specific contexts leading to improved predictions in those contexts (Rahimi et al. 2018). In regards to healthcare technologies and the adoption of IT in healthcare, TAM is regarded as the most important model used to identify the factors influencing the adoption of IT in the health system as well as patients' perceptions and behaviors (Garavand et al. 2016; Ahlan and Ahmad 2015).

Aside from the original TAM, many previous studies have investigated technology acceptance and user adoption of technology-related healthcare tools for the elderly population by incorporating new variables such as perceived risk, technical efficacy, and perceived cost, user enjoyment, facility conditions, and other variables including perceived irreplaceability, perceived credibility, compatibility, and social influence according to their contexts of the studies. Some studies explored the continued intention to use wearable devices by seniors while considering cognitive age and subjective well-being of the seniors (Ahmad et al. 2020; Farivar et al. 2020; Liu et al. 2022).

*2.3. Hypothesis Development*

2.3.1. Perceived Usefulness and Perceived Ease of Use

Perceived Usefulness (PU) refers to the degree to which a person believes that using a particular technology would improve his/her quality of life, and Perceived Ease of Use (PEOU) is defined as the extent to which a person believes that using a technology is free of effort (Davis 1989). The vast majority of studies conducted through the TAM approach assume that perceptions of usefulness are associated with the acceptance and usage behavior of particular products (Chen and Chan 2014a). Based on this, it is suggested that users will accept a new technology when their perceptions of the usefulness or advantage of its use are clearly positive. The TAM assumes that an individual's perception of how easy a technology is to engage with impacts their evaluation of its usefulness (Davis 1989).

From the study of the systematic review of factors influencing the adoption of ICT by healthcare professionals, Gagnon et al. (2012) concluded that PU of the system and PEOU

were the two most influential factors. These two factors are the main components of the original TAM. Therefore, these key variables will be explored in this study.

**H1.** *Perceived usefulness is positively related to attitude toward using digital health wearables.*

**H2.** *Perceived ease of use is positively related to attitude toward using digital health wearables.*

### 2.3.2. Social Influence/Social Impact

Social influence is defined as the degree to which others (family, friends, peers, and caregivers) believe (either positive or negative) something will affect someone to use the new technologies in healthcare (Venkatesh et al. 2003; Venkatesh et al. 2012; Andersen 1995). Many studies previously showed the importance of social influence in the decision of using technology for aging in place and social influence could come from children, family, peers, and professional caregivers for older adults (Lorenzen-Huber et al. 2011; Demiris et al. 2008; Porter and Ganong 2002).

In a study by Holtz and Krein (2011), social impact is found to be the most important factor in the adoption of electronic patient records. In similar studies, social impact is shown to be an important factor in the decision to use electronic medical records and the factor affecting the adoption of health information technology (Wills et al. 2008; Kijsanayotin et al. 2009). However, a study by Cohen et al. (2013) showed that social impacts had less importance and influence in South Africa on the implementation of electronic prescribing. Based on the findings of previous studies, it is hypothesized that:

**H3.** *Social Influence is positively related to attitude toward using digital health wearables.*

### 2.3.3. Facilitating Conditions

Facilitating conditions refer to which extent people believe that an organizational and technical infrastructure exists to support the system and motivate an individual to use that information system (Venkatesh et al. 2003). According to previous studies, it is shown that found that facilitating conditions such as training programs, technical support, and financial aid directly contribute to the intention to use smart wearables and have a positive effect on their use of innovative technologies by helping them overcome the concern towards the use (Mitzner et al. 2010; Lee and Coughlin 2015).

Venkatesh et al. (2003) found that facilitating conditions without adding any moderator is not significant to predict intention to use the system when the construct of effort expectancy is used in the same model, but rather when it is moderated by age and experience; it had a strong effect for older workers with increasing experience. Facilitating conditions such as access, cost, and availability of technical support are also considered important for older adults (Pan and Jordan-Marsh 2010). Therefore, it is hypothesized that:

**H4.** *Facilitating condition is positively related to attitude toward using digital health wearables.*

### 2.3.4. Attitude towards Using

Attitudinal factors (ATT) were defined as "an individual's positive or negative feelings or appraisal about using gerontechnology" (Ajzen 2015; Venkatesh et al. 2003). Previous studies showed that the elderly who have a higher positive attitude toward technology tend to use the technology. However, in the study by Chen and Chan (2014b), the attitude was not significant towards technology usage, which indicates that the elderly might not intend to use healthcare technology although they have a positive attitude towards the relevant technology. In this study, attitude will be measured as one of the factors influencing the behavioral intention of the elderly to use healthcare wearables. Therefore, it is hypothesized that:

**H5.** *Attitudes toward use are positively related to behavioral intentions to use digital health wearables.*

2.3.5. Behavioral Intention to Use

One of the key components of TAM is Behavioral Intention (BI) to use or adopt a technology which is influenced by one's attitude toward using the technology. In the original TAM, BI is defined as "the degree to which an individual has formulated conscious plans to perform or not perform some specified behavior in future," and it is proven that one's behavioral intention to use technology positively affects their actual usage of technology (Davis 1989; Venkatesh et al. 2003). Therefore, many studies focused on behavioral intention to use technology to predict the actual use and technology acceptance (Chen and Chan 2014b; Deng et al. 2014; Hoque and Sorwar 2017; Li et al. 2019).

Moreover, perceived usefulness is specified to have an independent effect on Behavioral Intention to use (BI). Some studies reported that the relationship between Perceived Usefulness (PU) and intention to use or the actual use of health IT is statistically significant (Rawstorne et al. 2000; Liang et al. 2003; Liu and Ma 2006), but in some studies, the PEOU-BI relationship is found to be not statistically significant (Yi et al. 2006). According to the TAM, one's behavioral intention to use technology, attitude towards use, and actual use are influenced by two main predictors: Perceived Usefulness and Perceived ease of use.

Previous studies have proven that the user's intention to use technology has a positive effect on their actual usage of the technology. Hence, in this study, behavioral intention is identified as a dependent variable to predict the actual use of technology.

**3. Methodology**

In a study conducted by (Holden and Karsh 2010), it is mentioned that TAM is the most common and widely used model in the healthcare domain and some TAM relationships were consistently found to be significant. Several studies added and modified variables to enhance the predictive power of the model according to the relevant context for better benefit.

Since the TAM is developed on individual beliefs, it has a limitation in that social impact and subjective norms are ignored. Therefore, the original TAM with only two predictive variables known as PU and PEOU would not be adequate to explain the elderly's technology acceptance in a healthcare context. Due to this, our study will use the extended TAM by including two more variables to understand the technology acceptance of Korean seniors and their intention to use digital health wearables (e.g., health smart watches). In addition to "Perceived Usefulness" and "Perceived Ease of Use" found in the original TAM, new variables such as "Social Impact" and "Facilitating Condition" are incorporated into the model. Therefore, Figure 1 shows the framework of this research based on the hypotheses.

*3.1. Data Collection*

This study adopted the quantitative research method to test the proposed model and conducted a cross-sectional questionnaire survey from October 2022 to November 2022. The data were collected by both paper-based and online-based (google form) surveys and distributed mainly to the South Korean elderly living in Busan. For sample selection, the total size of the older-adult population of Korea was considered as 8,151,867 in 2020; performing an equation for an infinite population (more than 100,000 people) with a confidence level of 99% and a margin of error of 7.5%, the result was 170, which is used as a sample size for this study. The selection process of the respondents was by randomization, and the survey was conducted by convenience sampling. One of the questions in the survey was about whether they had ever used or known of digital health wearables, and the respondents who answered "No" to this question were excluded from the study. However, due to the difficulty of collecting information, the study required facilitators who helped the elderly to answer the online survey form whereas most of the respondents filled out the paper-based questionnaire. The responses mainly came from elderly healthcare institutions and churches in Busan. Furthermore, many studies have defined "older adults" differently by various researchers across different countries, and most of them accepted older adults as being 60–65 years of age or older (Chen and Chan 2011). In this study, older adults that are

56 years or older are considered as accepted participants following past studies on aging and technology use (Palmiero et al. 2016; Hanafi et al. 2020; Wu and Song 2021).

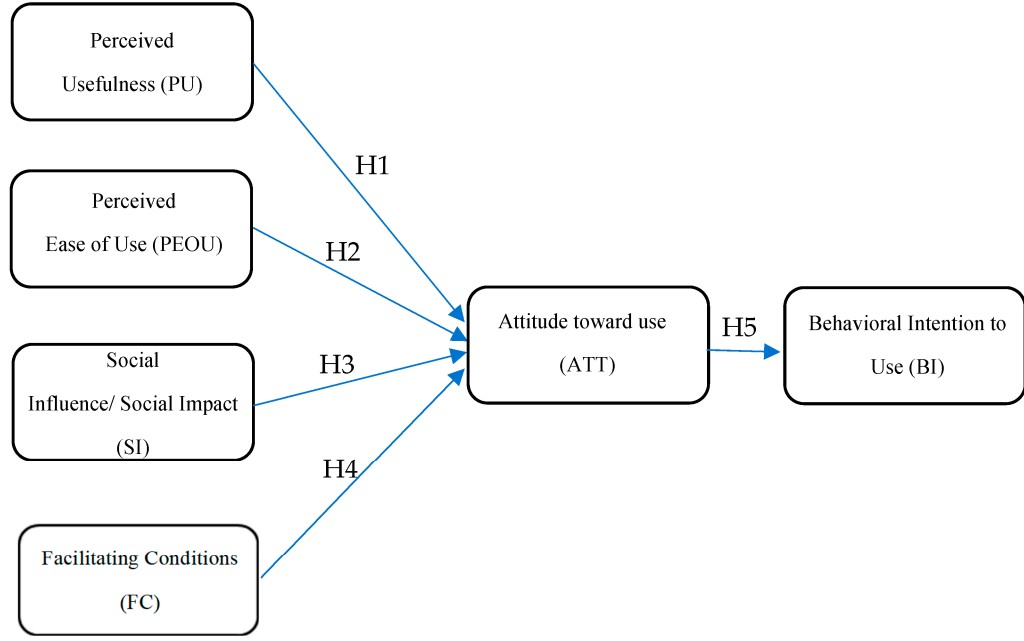

**Figure 1.** Research Model.

### 3.2. Measurement Instruments

The questionnaire was originally developed and structured in English and then translated into Korean by a native Korean speaker. Also, back translation is done to ascertain that the content and original meaning of all items and the expressions are preserved and consistent in both versions.

All measurement items for each variable in the model were developed and adapted from previous studies to suit the context of digital health wearables as shown in Table 1. The questionnaire was divided into three parts. The first part pertained to the general description of a healthcare smartwatch and a brief explanation of its use to help detect one's health status.

The second part included questions on sociodemographic information of respondents, mainly regarding age, gender, occupation, number of co-habitants in a household, income, and expense information. The third part consisted of the question about the type of chronic diseases the respondents have; some respondents reported having more than one type of chronic disease. All questions related to the six variables that are included in our proposed model were accessed in this part as well. Variables such as perceived usefulness, perceived ease of use, and intention to use digital health wearables were measured by three items each and were adapted from measurement items published by Venkatesh and Davis (2000). Social influence and Facilitating conditions were measured using three items each using measurement items published by Venkatesh et al. (2011, 2003). Finally, to measure attitude toward use, three items were adapted from Ajzen and Venkatesh et al. (Ajzen 2015; Venkatesh et al. 2003). We used the five-point Likert scale ranging from strongly disagree to strongly agree to measure all the constructs in the research model. The responses were recorded on a scale from Strongly disagree (1) to strongly agree (5) for data analysis.

**Table 1.** Measurement items of constructs.

| Construct | Item | Measurements | Reference |
|---|---|---|---|
| Perceived Usefulness | PU1 | Health Care Using a smart watch will help me manage my health. | (Venkatesh and Davis 2000) |
| | PU2 | I believe that using a health care smart watch will make my daily life safer. | |
| | PU3 | Using a health care smart watch will improve my quality of life. | |
| Perceived Ease of Use | PEOU1 | I think using a health care smart watch would be simple. | (Venkatesh and Davis 2000) |
| | PEOU2 | Learning to use a health care smart watch will be easy. | |
| | PEOU3 | Health care smart watch will be convenient to use. | |
| Social impact or Influence | SI1 | Family will approve of my use of a health care smart watch. | (Venkatesh et al. 2011, 2003; Ahmad et al. 2020) |
| | SI2 | Acquaintances will recommend that I use a health care smart watch. | |
| | SI3 | Acquaintances will approve of me using a health care smart watch. | |
| Facilitating conditions | FC1 | I will know how to use the health care smart watch. | (Venkatesh et al. 2011, 2003; Liu et al. 2022) |
| | FC2 | If I encounter difficulties using your health care smart watch, I think someone will be able to help. | |
| | FC3 | I have sufficient financial resources to use a health care smart watch. | |
| Attitude towards use | ATT1 | Using a health care smart watch will have a positive impact on my life | (Ajzen 2015; Venkatesh et al. 2003) |
| | ATT2 | Using a health care smart watch will benefit my health. | |
| | ATT3 | I have positive thoughts about health care smart watches. | |
| Behavioral Intention to Use | BI1 | I would use a health care smart watch if given the opportunity. | (Venkatesh and Davis 2000) |
| | BI2 | I will use a health care smart watch for my health care. | |
| | BI3 | I will use a health care smart watch to change my life for the better. | |

### 3.3. Data Analysis Procedure

The research model was validated using the partial least squares (PLS) method, which is based on structural equation modeling. First, we used measurement analysis, including factor loading, the average variance extracted (AVE), Cronbach alpha, and path coefficient to assess the validity and internal consistency of our research model, as well as to verify the correlation between different variables. SmartPLS 3 (SmartPLS GmbH, Oststeinbek Germany) was employed in our study to analyze the collected data.

## 4. Results

### 4.1. Demographic Characteristics

A total of 170 questionnaires were collected, and Table 2 shows demographic results. In this study, 75 respondents were male (44.1%) and 95 were female (55.9%).

**Table 2.** Demographics of respondents (N = 170).

| | | Frequency (N = 170) | Percent | Mean | SD |
|---|---|---|---|---|---|
| Age | 56~65 | 55 | 32.4 | 1.84 | 0.685 |
| | 66~75 | 89 | 52.4 | | |
| | 76~85 | 25 | 14.7 | | |
| | 86~95 | 1 | 0.6 | | |
| Gender | Male | 75 | 44.1 | 1.56 | 0.498 |
| | Female | 95 | 55.9 | | |
| Occupation | Office worker | 18 | 10.6 | | |
| | Professional | 18 | 10.6 | 3.90 | 1.454 |
| | Self-employed | 26 | 15.3 | | |
| | Public official | 9 | 5.3 | | |
| | Dependent | 99 | 58.2 | | |
| Number of people living together with | 1 | 12 | 7.1 | 2.84 | 0.965 |
| | 2 | 59 | 34.7 | | |
| | 3 | 44 | 25.9 | | |
| | 4 and above | 55 | 32.4 | | |
| Income | <500,000 won | 13 | 7.6 | 3.33 | 1.249 |
| | 500,000~ 1 million | 34 | 20 | | |
| | 1–2 million | 47 | 27.6 | | |
| | 2–3 million | 36 | 21.2 | | |
| | >3 million | 40 | 23.5 | | |
| Expense | <500,000 won | 16 | 9.4 | | |
| | 500,000~ 1 million | 46 | 27.1 | 3.02 | 1.174 |
| | 1–2 million | 48 | 28.2 | | |
| | 2–3 million | 39 | 22.9 | | |
| | >3 million | 21 | 12.4 | | |
| * Chronic diseases | No disease | 31 | 18.2 | | |
| | Hypertension | 63 | 37.1 | | |
| | Heart disease | 31 | 18.2 | | |
| | Diabetes | 61 | 35.9 | | |
| | Gastrointestinal | 28 | 16.5 | | |
| | Arthritis | 44 | 25.9 | | |
| | Dementia | 11 | 6.5 | | |
| | Dyslipidemia | 6 | 3.5 | | |

* Notes: Some respondents are reported to have more than one chronic disease.

Over half of the respondents (52.4%) are between the ages of 66–75, and 32.4% of the respondents are aged 56–65 years old, followed by respondents aged 76 to 85 years old (14.7%), while those aged 86–95 years old represented 0.6% of the total sample. As for the occupation of the respondents, both office workers and professionals each represented (10.6%) of the total respondents, whereas 15.3% and 5.3% of the sample were self-employed and public officials, respectively.

The majority of the respondents (34.7%) were living in a two-person household, 32.4% in household living four person households or above, 25.9% in three persons households, followed by those (7.1%) living in one person households. As for the income level, 27.6% of respondents have monthly income of 1–2 million won, 23.5% higher than 3 million won, 21.2% with 2–3 million won, followed by 20% with monthly income of 500,000–1 million won, and 7.6% of respondents with income less than 500,000 won. According to monthly expense level, 28.2% are with 1–2 million won monthly expense, 27.1% with expenses of 500,000–1 million won, 22.9% with 2–3 million, 12.4% of respondents with expenses higher than 3 million, and 9.4% with less than 500,000 won of monthly expenses. As for chronic diseases, the majority (37.1%) of the respondents have hypertension whereas a few (3.5%) of the respondents have Dyslipidemia. Some of the respondents reported to have more than one chronic disease. Table 2 presents the demographic characteristics of the respondents.

### 4.2. Measurement Model

The measurement model's accuracy and convergence validity were assessed. The Cronbach's alpha test is used to confirm internal reliability, and an internal consistency indicator of >0.7 is considered appropriate. Table 3 contains a list of the loadings, SMC, composition reliability, AVE, and Cronbach's alpha. Convergent validity was confirmed by the average variance extracted (AVE) values and composite reliability coefficients (CR), both of which were more than 0.60 (Bagozzi and Yi 1991). Our variables' Cronbach's alpha values demonstrate that they are internally reliable, and their loading and AVE values match the requirements for convergence validity. Cross-validation is a resampling procedure used to obtain nearly unbiased estimates of model performance without sacrificing the sample size. Tables 3–5 present the model validation results at each of the points of measurement.

**Table 3.** Factor loading and Reliability Test.

| Constructs | Item | Loading | VIF | SMC | Composite Reliability | AVE | Cronbach's α |
|---|---|---|---|---|---|---|---|
| Perceived Usefulness | PU1 | 0.896 | 2.367 | 0.803 | | | |
| | PU2 | 0.904 | 2.617 | 0.817 | 0.926 | 0.808 | 0.881 |
| | PU3 | 0.896 | 2.400 | 0.803 | | | |
| Perceived Ease of Use | PEOU1 | 0.924 | 3.335 | 0.854 | | | |
| | PEOU2 | 0.934 | 3.764 | 0.872 | 0.939 | 0.838 | 0.903 |
| | PEOU3 | 0.887 | 2.351 | 0.787 | | | |
| Social Influence or Impact | S1 | 0.892 | 2.318 | 5.373 | | | |
| | S2 | 0.872 | 2.178 | 4.744 | 0.922 | 0.798 | 0.874 |
| | S3 | 0.916 | 2.746 | 7.541 | | | |
| Facilitating Condition | FC1 | 0.890 | 2.156 | 0.792 | | | |
| | FC2 | 0.899 | 2.454 | 0.808 | 0.918 | 0.788 | 0.865 |
| | FC3 | 0.874 | 2.175 | 0.764 | | | |
| Attitude towards use | ATT1 | 0.894 | 2.295 | 0.799 | | | |
| | ATT2 | 0.893 | 2.375 | 0.797 | 0.922 | 0.797 | 0.873 |
| | ATT3 | 0.891 | 2.328 | 0.794 | | | |
| Behavioral Intention to use | BI1 | 0.899 | 2.492 | 0.808 | | | |
| | BI2 | 0.922 | 3.005 | 0.850 | 0.930 | 0.817 | 0.888 |
| | BI3 | 0.890 | 2.403 | 0.792 | | | |

**Table 4.** Discriminant Validity (Fornell-Larcker).

| Items | PU | PEOU | SI | FC | ATT | BI |
|---|---|---|---|---|---|---|
| PU | **0.899** | | | | | |
| PEOU | 0.664 | **0.915** | | | | |
| SI | 0.758 | 0.583 | **0.893** | | | |
| FC | 0.663 | 0.784 | 0.617 | **0.888** | | |
| ATT | 0.768 | 0.701 | 0.644 | 0.704 | **0.893** | |
| BI | 0.692 | 0.682 | 0.644 | 0.647 | 0.716 | **0.904** |

Note: Values in bold represent the square root of the variance extracted (AVE) and the values outside the diagonal represent the correlations between the constructs. PEOU: Perceived ease of use; PU: Perceived usefulness; SI: Social influence; FC: Facilitating condition; ATT: Attitude towards use; BI: Behavioral intention to use.

If a factor's discriminant validity is to be regarded as significant, its square root of average variance for each latent variable must be higher than the other correlation coefficient. The detailed cross-loading values can be seen in Table 5 in which each construct item placed a greater emphasis on it than any other construct which were shown as bolded values in Table 5. The study's findings indicate that the requirements for discriminant validity are met. Tables 3–5 present the results of our model's validity and reliability.

**Table 5.** Discriminant Validity (Cross-Loadings).

| Items | PU | PEOU | SI | FC | ATT | BI |
|---|---|---|---|---|---|---|
| PU1 | **0.896** | 0.574 | 0.711 | 0.610 | 0.705 | 0.647 |
| PU2 | **0.904** | 0.629 | 0.675 | 0.606 | 0.670 | 0.618 |
| PU3 | **0.896** | 0.588 | 0.657 | 0.572 | 0.693 | 0.601 |
| PEOU1 | 0.641 | **0.924** | 0.572 | 0.757 | 0.666 | 0.638 |
| PEOU2 | 0.592 | **0.934** | 0.519 | 0.713 | 0.629 | 0.650 |
| PEOU3 | 0.588 | **0.887** | 0.508 | 0.681 | 0.629 | 0.562 |
| SI1 | 0.689 | 0.518 | **0.892** | 0.562 | 0.608 | 0.543 |
| SI2 | 0.647 | 0.489 | **0.872** | 0.520 | 0.555 | 0.619 |
| SI3 | 0.695 | 0.554 | **0.916** | 0.571 | 0.609 | 0.576 |
| FC1 | 0.621 | 0.724 | 0.533 | **0.890** | 0.667 | 0.629 |
| FC2 | 0.561 | 0.656 | 0.537 | **0.899** | 0.618 | 0.528 |
| FC3 | 0.582 | 0.706 | 0.576 | **0.874** | 0.586 | 0.562 |
| ATT1 | 0.717 | 0.628 | 0.621 | 0.620 | **0.894** | 0.666 |
| ATT2 | 0.654 | 0.604 | 0.585 | 0.650 | **0.893** | 0.624 |
| ATT3 | 0.684 | 0.646 | 0.566 | 0.617 | **0.891** | 0.627 |
| BI1 | 0.644 | 0.659 | 0.627 | 0.621 | 0.655 | **0.899** |
| BI2 | 0.642 | 0.612 | 0.588 | 0.586 | 0.657 | **0.922** |
| BI3 | 0.589 | 0.577 | 0.529 | 0.547 | 0.629 | **0.890** |

*4.3. Hypothesis Testing*

To test our hypotheses, we employed a bootstrapping method (5000 times) of the Smart PLS 3 to calculate the path coefficients and t values. The bootstrap method was used to confirm the stability of data by randomly choosing subsamples from the observed data. Four variables (perceived usefulness, perceived ease of use, social influence, and facilitating condition) predict 67.6% of the attitude ($R^2 = 0.676$), which predicts 51.3% of the behavior intention ($R^2 = 0.513$). The relationship between variables was tested by using path coefficient ($\beta$) and t statistics. The PLS results for the hypotheses are shown in Table 6. The results indicate that all hypotheses, except H3, were supported. Both the perceived usefulness and the perceived ease of use demonstrated the positive significant effect on attitude toward the use of a smart healthcare watch (H1: $\beta = 0.425$, $p < 0.05$; H2: $\beta = 0.204$, $p < 0.001$). Meanwhile, social influence had no significant effect on attitude towards the use of a healthcare smart watch ($\beta = 0.095$, $p > 0.05$) and thus H3 was rejected. Facilitating condition had a significant positive impact on the attitude towards use ($\beta = 0.204$, $p < 0.05$), thus H4 was supported. And lastly, the attitude toward telehealth had a significantly positive influence on the intention to use telehealth ($\beta = 0.716$, $p < 0.001$).

**Table 6.** Hypothesis analysis results.

| Hypothesis | Path | | β | *t*-Value | Comments |
|---|---|---|---|---|---|
| H1 | PU | → ATT | 0.425 | 4.296 *** | Supported |
| H2 | PEOU | → ATT | 0.204 | 2.136 * | Supported |
| H3 | SI | → ATT | 0.095 | 1.285 | Not supported |
| H4 | FC | → ATT | 0.204 | 2.276 * | Supported |
| H5 | ATT | → BI | 0.716 | 18.047 *** | Supported |

PEOU: Perceived ease of use; PU: Perceived usefulness; SI: Social influence; FC: Facilitating condition; ATT: Attitude towards use; BI: Behavioral intention to use, $p < 0.05$ *, $p < 0.001$ ***.

**5. Discussion and Conclusions**

The main goal of this study was to investigate the factors that influence Korean older adults' acceptance and behavioral intention to use digital health wearables, especially smart healthcare watches. Based on the original constructs of TAM which are perceived usefulness and perceived ease of use, our study incorporated two additional constructs, social influence and facilitating condition. In this study targeting the Korean elderly population, we confirmed that perceived usefulness, ease of use, and facilitating conditions

are the important factors that have positive significant effect on attitude toward use of smart healthcare wearables.

It was found that the original TAM constructs of perceived usefulness and perceived ease of use had positive influence on attitude towards use of digital health wearables. In this study, perceived usefulness ($\beta = 0.425$) had significant influence on the attitude of Korean elderly people on smart healthcare watches. Similarly, perceived ease of use ($\beta = 0.204$) had a positive impact on the attitude towards use of wearable technology among Korean elderly people. These findings are consistent with the findings of previous studies (Deng et al. 2014; Li et al. 2019; Tural et al. 2020).

The results demonstrated that facilitating conditions ($\beta = 0.204$) were positively related to the attitude of Korean elderly people towards the use of digital healthcare wearables, whereas social influence ($\beta = 0.095$) was found not to be a significant factor in forming the positive attitude of the Korean elderly population to use wearable technologies. These results aligned with findings of previous studies (Chen and Chan 2014b; Deng et al. 2014; Li et al. 2019; Macedo 2017; Yein and Pal 2021). The first two findings related to perceived usefulness and perceived ease of use imply that digital health wearables need to meet the expectation of Korean elderly people in terms of simplicity in usage and the practical usefulness in their lives in order to form positive attitudes towards these technologies and encourage their use. In addition, the third finding suggests that social influence is not that important for Korean elderly people to form a positive attitude towards use of digital wearables. Their friends, family, and acquaintances will not influence greatly whether the Korean elderly will intend to use these wearables unless they themselves are willing to use those technologies.

The fourth finding regarding facilitating conditions indicates that Korean elderly people will likely use digital healthcare wearables if they receive environmental support and guidelines whenever they need help in using digital healthcare wearables. Lastly, in consistency with previous studies (Chen et al. 2021; Deng et al. 2014; Tural et al. 2020), it is found that the attitudes of the Korean elderly had a significant effect on their behavioral intention to use digital healthcare wearables. These showed that the elderly who have a higher positive attitude towards the use of these technologies will have a higher behavioral intention to use smart healthcare watches.

### 5.1. Theoretical Implications

This study provides further evidence of the role of perceived usefulness, perceived ease of use, social influence, and facilitating conditions in Korean older adults' attitudes and their behavioral intention to use digital healthcare wearables. This paper also explores the effects of the attitude of the Korean elderly towards use on their behavioral intention to use digital health wearables by using the extended Technology Acceptance Model.

This study further confirms and validates the credibility of TAM and demonstrates the significance of two antecedents in the original TAM for attitude toward the use of digital health technologies and their behavioral intention to use. The findings of this study regarding the significance of Perceived usefulness and Perceived Ease of Use are consistent with the findings of previous studies (Yang et al. 2016; Or et al. 2011; Davis et al. 1989). However, in some studies, it is reported that the relationship between perceived ease of use and attitude towards the use of digital health and wearable technologies is not statistically significant (Lazaro et al. 2020; Hu et al. 1999).

Social influence is found to be not significant in this study in relation to the attitude of Korean elderly people towards their use of digital health. Contradicting this study's result, social influence was shown as an important factor for older adults to adopt mobile health and wearable technologies (Lazaro et al. 2020; Kijsanayotin et al. 2009; Lee and Coughlin 2015; Hoque and Sorwar 2017). The result of our study also indicated that facilitating conditions such as providing training programs, financial aid, and technical support are positively related to attitudes toward using digital health wearables, which is consistent with prior studies (Mitzner et al. 2010; Lee and Coughlin 2015; Pan and Jordan-Marsh

2010). The finding of this study showed a positive correction between the attitude towards the use of digital health wearables and their behavioral intention to use digital healthcare wearables.

*5.2. Practical Implications*

The results from this study can be applied and utilized by digital healthcare wearables developers, elderly healthcare services, medical equipment manufacturers, promoters, and health practitioners in creating practical and value-added healthcare smartwatches designed specifically for elderly people. Based on findings from this study, the developers and manufacturers must be aware that the features and functions included in healthcare smartwatches should be simple and easy to use while providing practical usefulness in the daily lives of Korean elderly people which will increase their intention to use healthcare wearables.

Moreover, the availability of technical support and help is needed, and the usage should be explained clearly and well in advance to the Korean elderly in order to encourage them to use healthcare smartwatches. According to the result of this study, the Korean elderly will not be greatly influenced by their family and acquaintances in using digital healthcare wearables. Regardless of influences from their social circles, they will not intend to use these technologies unless they perceive these digital wearables are useful, easy to use and given necessary technical support whenever they need while using them.

In addition, the results show that the Korean elderly will consider if they have sufficient financial resources to use particular technology which suggests that manufacturers and suppliers should make these healthcare smart watches affordable and pricing should be based on the average income of Korean elderly people. Finally, the manufacturers, marketers, and suppliers need to increase awareness of the Korean elderly about the benefits of digital health wearables and should attempt to create positive perception among Korean elderly people by listening to their feedback and modifying product features to meet their needs. This will help create positive attitudes in the Korean elderly towards the use of digital healthcare wearables, and the more they perceive these technologies positively, the higher their intention to use these wearable technologies.

## 6. Limitations and Directions for Future Research

This study is not free from limitations and has its shortcomings. Since the data were collected only in Busan, South Korea, the significance of the results of the study is limited. Plus, the sample size was relatively small considering the large number of elderly populations across Republic of Korea. In the future, a larger sample size is recommended for data collection. In addition, the concept of "older adult" is universal and aging can be viewed differently, not only in the age's number but also based on other factors as well. Moreover, to provide a deeper understanding of this study, the future researchers are recommended to consider additional variables, such as age-related characteristics, physical changes, digital literacy, technically savvy- technology experience, and adoption in order to present a wider perspective and more conclusive view on behavioral intention of the Korean elderly people to use digital healthcare wearables.

**Author Contributions:** Conceptualization, K.S.L.T.Z. and H.-S.K.; methodology, K.S.L.T.Z. and S.K.; analysis, K.S.L.T.Z., S.K. and H.-S.K.; writing—original draft preparation, K.S.L.T.Z., S.K.; writing— review and editing, K.S.L.T.Z., S.K. and I.F.F.; supervision and funding acquisition, H.-S.K. All authors have read and agreed to the published version of the manuscript.

**Funding:** This research was financially supported by the Ministry of Trade, Industry and Energy, Korea, under the "World Class Plus Program (R&D, P0020673)" supervised by the Korea Institute for Advancement of Technology (KIAT).

**Institutional Review Board Statement:** Not applicable.

**Informed Consent Statement:** Not applicable. This study does not involve ethics of humans or animals. The research poses no risk to subjects and the results will not adversely affect the rights and welfare of the subjects who are involved in the research.

**Data Availability Statement:** Data available in a publicly accessible repository.

**Conflicts of Interest:** The authors declare no conflict of interest.

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
