# Peer review of "A Study on Technology Acceptance of Digital Healthcare among Older Korean Adults Using Extended Tam (Extended Technology Acceptance Model)"

_admsci, doi:10.3390/admsci13020042_

Round 1

Reviewer 1 Report

Thank you for giving me the opportunity to review this manuscript entitled “A study on technology acceptance of digital healthcare among older Korean adults using extended TAM (extended Technology acceptance model).

1.     Overall

This study is very valuable because there are very few cases (studies) of Korean older adults and the usage of technology for their health care. Moreover, it is very difficult to recruit the respondents.

 2.     Literature review

In figure 1, the hypotheses need to be presented.

 3.     Literature review

The “extended TAM” should be more highlighted.

 4.     Method: data collection

This study do not need to explain like

“ But in this study, older adults that are 56 343 years or older are considered as accepted participants following past studies on aging and 344 the technology use of older adults in which different age cohorts are comparatively stud- 345 ied (young adults: 18–35 years; middle-aged adults: 36–55 years; older adults: 56 or more 346 years) (Palmiero, Nori, and Piccardi 2016; Hanafi et al. 2020; Wu and Song 2021).”

5.     Method: the data collection and data procedures

The contents of the data collection and data procedures should be sufficiently explained.

What is a sampling technique?

How do this study recruit older adults who use health care via smart watch?

How do all older adults easily have opportunity to access to this survey via online? Because data collection from older adults are very difficult because they are not very patients or confident when they fill out the online survey.

 6.     method

The measurement items come from mainly one or two research articles. The contribution of the suggesting new constructs failed in this study. The contributions of this model should be highlighted.

 7.      Minor

from Strong disagree (1) to strongly agree (5)

è Strong -> strongly

 8.     Results

The order of the constructs are very messy.

I prefer to see the consistent order of the hypotheses and the constructs in all tables.

Please check Tables 1, 3, and 4.

 9.     Results

Table 4 shows the discriminant validity, however, the correlation values between constructs seem to be high. How can the authors insist that discriminant validity is confirmed?

10. results

4.3. hypothesis testing is written in bold letters. Please overall revise the manuscript.

11. Results

The authors should report standardized regression coefficients instead of t-value. Because t-value give readers information about statistically significant levels. Also, if you can, please mark stars (*) to make readers easily recognize the significance of the results.

12. Minor

The format of the reference style is not following ACS. The authors need to revise the format for MDPI.

Reviewer 2 Report

The topic of this paper is interesting, and with potential to dive deep inside, however, I have some comments and recommendations to be addressed as follows:

The abstract would benefit from brief practical implications.

Consistent English language proofing needs to be performed.

Section 1 Please revisit the introduction to include more emphasis on the most significant research papers in the research area that need to be relevant for explaining the keywords from the objective would be advantageous to the reader. 

The last paragraph in the introduction should introduce the rest of the paper. 

Section 2 Related literature presents a smooth structure with some interesting ideas; however, it would be better if supported with a more recent literature review.

Section 3 Methodology: Is 170 participants enough to generalise your results? Justify it, please

Section 4 Results can be analysed and explained further.

Section 5 Conclusion can be enhanced if the implications are explained further as well as limitations and further studies can be highlighted. 

Round 2

Reviewer 1 Report

Thank you for your revision. 

I have very minor comments for this manuscript. 

1. One sentence is a paragraph in the manuscript. Please read and revise the manuscript overall. 

2. In table 2, title should be above the content. The number of the respondents should be matched in the table and on the title. 

3.  R-squared values seem not to be reported. 

4. the mediation effect is also missing. 
